# Pharmacological blood pressure control and outcomes in patients with hypertensive crisis discharged from the emergency department

Yu-Ting Lin[1], Yen-Hung Liu[2], Ya-Luan Hsiao[3], Hsiu-Yin Chiang[1], Pei-Shan Chen[1], Shih-Ni Chang[1], Hsiu-Chen Tsai[1], Chun-Hung Chen[2], Chin-Chi Kuo[1,4]*

1 Big Data Center, China Medical University Hospital, Taichung, Taiwan, 2 Department of Emergency Medicine, China Medical University Hospital, Taichung, Taiwan, 3 Department of Health Policy and Management, Johns Hopkins University Bloomberg School of Public Health, Baltimore, Maryland, United States of America, 4 Division of Nephrology, Department of Internal Medicine, China Medical University Hospital, Taichung, Taiwan

* chinchik@gmail.com

## Abstract

Pharmacological blood pressure (BP) intervention for high blood pressure is controversial for a wide spectrum of hypertensive crisis in the emergency department (ED). We evaluated whether medical control of BP altered the short- and long-term outcomes among patients with hypertensive crisis who were discharged from the ED under universal health care. This retrospective cohort comprised 22 906 adults discharged from the ED of a tertiary hospital with initial systolic BP $\geq$ 180 mmHg or diastolic BP $\geq$ 120 mmHg between 2010 and 2016. The main exposure was the use of antihypertensive medication during the ED stay. Clinical endpoints were revisits to the ED or inpatient admission (at 7, 30, and 60 days), cardiovascular mortality (at 1, 3, and 5 years), and incident stroke (at 1, 3, and 5 years). The associations between pharmacological intervention for BP and outcomes were evaluated using multivariable Cox proportional-hazards models. Of the patient data analyzed, 72.2% were not treated pharmacologically and 68.4% underwent evaluation of end-organ damage. Pharmacological intervention for BP was significantly associated with a 11% and 11% reduced risk of hospital revisits within 30 or 60 days of discharge from ED, respectively, particularly among patients with polypharmacy. No association between pharmacological intervention for BP and incident stroke and cardiovascular mortality was observed. A revision of diagnostic criteria for hypertensive crisis is essential. Although pharmacological intervention for BP may not alter the long-term risk of cardiovascular mortality, it significantly reduces short-term health care utilization.

## Introduction

Hypertensive crisis (HTN-C), defined as systolic blood pressure (SBP) $\geq$ 180 mmHg or diastolic blood pressure (DBP) $\geq$ 120 mmHg, may lead to progressive end-organ damage and even devastating clinical consequences such as stroke, myocardial infarction, and renal failure

**Data Availability Statement:** Data cannot be shared publicly in order to protect patient confidentiality. Data are available from the China

 

Medical University Hospital Institutional Data Access / Ethics Committee (contact via Ms. Maggie Shih; Email: a6034@mail.cmuh.org.tw) to researchers who meet the criteria for access to confidential data.

**Funding:** This study was supported by the Ministry of Science and Technology of Taiwan (grant number: 108-2314-B-039 -038 -MY3 and 109-2321-B-468 -001 -) and China Medical University Hospital, Taichung, Taiwan (grant number: CRS-106-018 and DMR-HHC-109-6). The funders had no role in study design, data collection and analysis, decision to publish, or preparation of the manuscript.

**Competing interests:** The authors have declared that no competing interests exist.

[1]. HTN-C can be further distinguished as hypertensive urgency (HTN-U) and emergency (HTN-E) based on the presence of end-organ damage. Patients with HTN-E exhibits signs of new or progressive injury of vital organs such as the brain, heart, and kidney. By contrast, patients with HTN-U are free of end-organ damage but may have non-life-threatening symptoms such as anxiety, headache, neck soreness, palpitations, and mild dyspnea [2]. A recent study reported that the prevalence of hypertensive crisis per 1000 visits to the emergency department (ED) has tripled from 1.8 to 4.6 in the United States [3]. In Taiwan, approximately one in every four Taiwanese has hypertension [4] and it is estimated that 1–2% of patients with hypertension may develop HTN-C [5]. Therefore, the management of hypertensive crisis and its associated complications are crucial in Taiwan.

For patients with hypertensive crisis, urgent reduction in BP is typically not required, except among patients with HTN-E [1,6]. The patient's target BP should not be set at a level much lower than the baseline BP [1]. In real-world practice, treatment of HTN-U has varied, ranging from prescribing medications for treating potential secondary causes such as pain or drug withdrawal to resting for at least 30 minutes [7]. A study found that among 379 patients who presented to the clinic with HTN-U, those referred to the ED and those sent home had similar rates of major adverse cardiovascular events [8]. Approximately 37.5% of those referred to the ED for HTN-U did not receive medical treatment for high BP [8]. However, existing evidence mostly compares the adverse events of medically treated HTN-U and HTN-E in the ED setting [9,10]. The short-term and long-term clinical outcomes such as ED revisits, cardiovascular mortality, and neurovascular emergencies in patients with the wide spectrum of HTN-C who do not receive antihypertensive treatment remain unclear [11]. To fill this knowledge gap, we conducted a large retrospective cohort study of ED-discharged patients presenting with HTN-C to the ED of a tertiary medical center. We compared the short- and long-term clinical outcomes between the discharged patients with HTN-C who received or did not receive antihypertensive agents during their ED stay and characterized the risk modifiers.

## Materials and methods

### Source population

This retrospective cohort study used the Clinical Research Data Repository (CRDR), which consolidates 14-year electronic medical records (EMRs) from China Medical University Hospital (CMUH). An average monthly volume of more than 12,000 patients has presented to the ED at CMUH since 2010 [12]. The source population consists of adult patients (age > 18 years) admitted to the ED between January 1, 2010, and December 31, 2016.

### Study population

Among the source population, patients with the first BP reading at an SBP ≥180 mmHg or a DBP ≥120 mmHg (i.e., HTN-C) and subsequently had at least one BP measurements during their ED stay were enrolled in this study [13]. BP measurements were excluded if the SBP was <50 mmHg or >270 mmHg or if the DBP was <30 mmHg or >160 mmHg. We included only the first episode that met the eligibility criteria if the patient had multiple ED visits that met the definition of HTN-C. Patients were excluded if they withdrew their registration from the ED, revisited the ED on the same day, died during ED admission, were hospitalized, discharged against the physician's advice, or referred to other institutions. The details of the selection process are described in Fig 1. The index date was the date of the first ED visit presenting with HTN-C. The study was approved by the Research Ethical Committee/Institutional Review Board of China Medical University Hospital (CMUH105-REC3-068), and the requirement of written informed consent was waived.

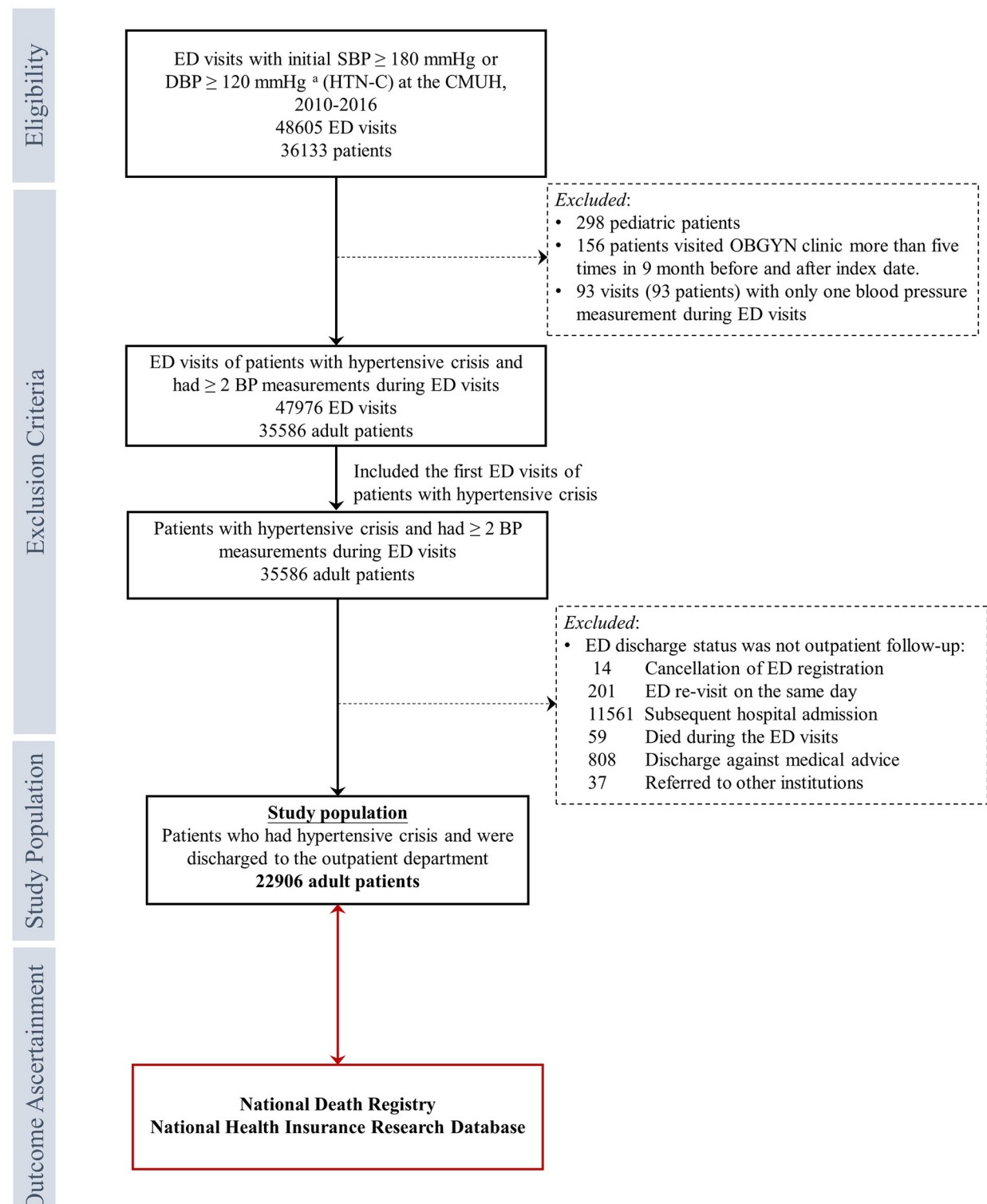

**Fig 1. Selection process of the study population.** Abbreviations: BP, blood pressure; CMUH, China Medical University Hospital; DBP, diastolic blood pressure; ED, emergency department; HTN-C, hypertensive crisis; SBP, systolic blood pressure. SBP measurements less than 50 mm/Hg or higher than 270 mm/Hg were excluded. DBP measurements less than 30 mm/Hg or higher than 160 mm/Hg were excluded.

## Exposure

Antihypertensive treatment prescribed during the index visit to the ED was defined as the use of angiotensin-converting enzyme inhibitors (ACEIs), angiotensin II receptor blockers (ARBs), calcium channel blockers (CCB), diuretics, α1 blockers, β blockers, imidazoline receptor agonists, hydrazinophthalazine derivatives, and organic nitrates (S1 Table).

## Covariables

EMR data obtained from CMUH-CRDR recorded within a 1-year window before the index date were used to compile the baseline comorbidities, relevant biochemical measures, and medication use. Indications of diabetes mellitus (DM) and chronic HTN were based on the clinical diagnosis of physicians using the International Classification of Disease, Revision 9, Clinical Modification diagnosis codes and/or on the use of glucose-lowering/antihypertensive agents within 1 year prior to the index date (S2 Table). A history of cardiovascular disease (CVD) was defined as the presence of coronary artery disease, myocardial infarction, or heart failure based on the documented ICD-9 diagnoses. A history of stroke was defined if patients had ever been registered in the National Catastrophic Illness Registry as having "cerebrovascular disease (acute stage)" prior to the index date. The National Catastrophic Illness Registry is regulated by the Ministry of Health and Welfare (MOHW) of Taiwan, and it currently covers 30 major disease categories, including stroke [14]. Serum creatinine levels at enrollment were used to define the baseline estimated glomerular filtration rate (eGFR) and the corresponding chronic kidney disease (CKD) status (eGFR < 60 mL/min/1.73 $m^2$). Examinations for suspected end-organ damage were defined as requesting a serum creatinine exam, troponin I exam, intravenous diuretic therapy, brain CT, or aortic dissection during the index ED visits, regardless of the exam results. BP variation was presented as slope change, which was estimated using a multilevel model, including a random intercept and a slope, with all available BP measurements clustered within the patients [15]. Polypharmacy was defined as receiving five or more unique medical prescriptions for at least 28 consecutive days per prescription within 1 year prior to the date of the index ED visit [16].

## Outcomes

The outcomes of interest were revisits to the ED or inpatient admissions within 7, 30, or 60 days following the index ED visit and CV mortality at 1, 3, and 5 years. The mortality data were obtained from the National Death Records from the Health and Welfare Data Science Center of the MOHW in Taiwan. The risk of incident stroke (at 1, 3, and 5 years) among patients without a history of stroke was also investigated.

## Statistical analyses

Continuous variables are presented as medians and interquartile ranges (IQRs), and they were analyzed using the Wilcoxon rank-sum test. Categorical variables are reported as frequency and proportions (%), and they were analyzed using the chi-square test or Fisher's exact test.

The associations between the exposure of antihypertensive treatment in the ED and the clinical outcomes of interest were estimated using multivariable Cox regression analysis. The time scale for survival analysis was the calendar date, and the late entry method was applied using index date as the individual entry time. Patients were followed up until the date of events or were censored at the corresponding observation time point (e.g., 7, 30, or 60 days after the index date). Multivariable Cox regression models were initially adjusted for demographic information and comorbidities, such as age, sex, diabetes, hypertension, CVD, and CKD;

subsequently adjusted for random slope of SBP, maximum SBP, and baseline eGFR; and finally adjusted for antiplatelet agents and polypharmacy.

Exploratory subgroup analysis was performed to evaluate the potential effect modification by age (<65 years vs. ≥65 years), sex, diabetes, hypertension, CKD (eGFR <60 mL/min/1.73 m² vs. ≥60 mL/min/1.73 m²), polypharmacy, and work-ups for suspected end-organ damage for all outcomes. We also provided the visualization information of subgroup analysis to show the association between pharmacological intervention for BP among patients with HTN-C stratified by subgroup and the effect modification. We performed sensitivity analyses including patients with (1) medication histories prescribed by other hospitals and (2) persistent high blood pressure above HTN-C criteria after the 4-hour ED stay. All statistical analyses were performed using SAS version 9.4 (SAS Institute Inc., Cary, NC, USA) and R version 3.5.1 (R Foundation for Statistical Computing, Vienna, Austria). The two-sided statistical significance level of $\alpha$ was set at 0.05.

## Results

### Clinical characteristics affecting decisions regarding pharmacological control of BP

Compared with discharged patients with HTN-C who did not receive any antihypertensive medication, older patients with more prevalent comorbidities such as DM, HTN, CVD, and stroke were more likely to receive pharmacological control for acute elevated BP (Table 1). Patients with antihypertensive medications were more likely surveyed for suspected end-organ damage and received brain CT scans, serum creatinine measurements, and troponin-I detection (Table 1). The median of the initial and maximum SBP was significantly higher in patients who received treatment for BP control compared with those who did not receive treatment, with a median difference of >10 mmHg, although the initial and maximum DBP were comparable between the two groups. The median time between the first and last BP measurements was also longer among patients with pharmacological control of BP than among patients without BP control (3.1 hours vs. 2.2 hours). Moreover, the slope of SBP and DBP reduction was significantly steeper in patients who did not receive antihypertensive medication (Table 1). The median SBP at discharge was significantly higher among patients who received antihypertensive medication than among those who did not receive antihypertensive medication (157 [QR 142–170] vs. 152 [IQR 136–165] mmHg, p-value < 0.001). Among patients with suspected end-organ damage, those prescribed with antihypertensive medications had a two-fold higher chance of undergoing tests, particularly the troponin-I test (Table 1). Patients who received antihypertensive medications in the ED for HTN-C were more likely to be chronically prescribed almost all antihypertensive medication classes, reflecting a higher prevalence of HTN (up to 86.6%) in this group (Table 1). Patients with pharmacological control of BP were more likely to receive multiple medications for chronic illnesses.

### Associations of pharmacological intervention for BP with short-term hospital revisits and long-term CV and neurological outcomes

Pharmacological intervention for BP was significantly associated with an 11% (95% confidence interval [CI], 3–18%) and 11% (95% CI, 4–18%) reduced risk of hospital revisits within 30 days or 60 days of discharge from the ED, respectively, but not with 7-day hospital revisits (adjusted hazard ratio [aHR], 0.94 [95% CI, 0.84–1.06]; Table 2). We did not observe an association between pharmacological intervention for BP among patients with HTN-C discharged from the ED setting and incidence of stroke and CV mortality at 1, 3, and 5 years (Table 2).

**Table 1. Baseline demographic and clinical characteristics of the study population.**

| Variables | Missing, n (%) | All patients | Pharmacological BP intervention | Non-pharmacological BP intervention | P-value[a] |
|---|---|---|---|---|---|
| N | | 22906 (100) | 6364 (27.8) | 16542 (77.2) | |
| **Age at ED admission (year)** | 0 (0) | 60.2 (48.5, 72.2) | 62.8 (52.5, 73.9) | 59.2 (46.8, 71.4) | < 0.001 |
| **Male, n (%)** | 0 (0) | 10516 (45.9) | 2767 (43.5) | 7749 (46.8) | < 0.001 |
| **Baseline comorbidities[b], n (%)** | | | | | |
| Diabetes | 0 (0) | 2496 (10.9) | 834 (13.1) | 1662 (10) | < 0.001 |
| Hypertension | 0 (0) | 10324 (45.1) | 4848 (76.2) | 5476 (33.1) | < 0.001 |
| Stroke | 0 (0) | 1878 (8.2) | 608 (9.6) | 1270 (7.7) | < 0.001 |
| Chronic kidney disease (eGFR<60 ml/min/1.73m$^2$) | 4739 (20.7) | 4621 (25.4) | 1669 (29.8) | 2952 (23.5) | < 0.001 |
| Cardiovascular disease | 0 (0) | 2070 (9) | 833 (13.1) | 1237 (7.5) | < 0.001 |
| **Exams for suspected EOD during the ED visit, n (%)** | | | | | |
| Workup for end-organ damage | 0 (0) | 15665 (68.4) | 5071 (79.7) | 10594 (64) | < 0.001 |
| Serum Creatinine | 0 (0) | 14696 (64.2) | 4837 (76) | 9859 (59.6) | < 0.001 |
| Troponin I | 0 (0) | 6352 (27.7) | 2947 (46.3) | 3405 (20.6) | < 0.001 |
| Brain CT scan | 0 (0) | 3330 (14.5) | 1084 (17) | 2246 (13.6) | < 0.001 |
| Intravenous diuretic | 0 (0) | 360 (1.6) | 356 (5.6) | 4 (0) | < 0.001 |
| Aortic dissection | 0 (0) | 9 (0) | 7 (0.1) | 2 (0) | 0.003 |
| **SBP measured during the ED visit, median (IQR)** | | | | | |
| Initial value (mmHg) | 0 (0) | 190 (183, 200) | 197 (187, 209) | 188 (182, 197) | < 0.001 |
| Second value (mmHg) | 0 (0) | 159 (143, 175) | 175 (155, 192) | 155 (140, 168) | < 0.001 |
| Maximum value (mmHg) | 0 (0) | 190 (183, 201) | 200 (189, 212) | 188 (182, 197) | < 0.001 |
| Slope[c] (mmHg/hour) | 0 (0) | -16.9 (-19.4, -13.5) | -13.8 (-17.6, -9.1) | -17.5 (-19.9, -15) | < 0.001 |
| Value at discharge (mmHg) | 0 (0) | 153 (138, 166) | 157 (142, 170) | 152 (136, 165) | < 0.001 |
| Value at discharge ≧180 mmHg, n(%) | 0 (0) | 1969 (8.6) | 824 (12.9) | 1145 (6.9) | < 0.001 |
| **DBP measured during the ED visit, median (IQR)** | | | | | |
| Initial value (mmHg) | 0 (0) | 106 (94, 120) | 106 (94, 120) | 106 (94, 120) | 0.593 |
| Second value (mmHg) | 0 (0) | 87 (78, 98) | 91 (80, 105) | 86 (77, 96) | < 0.001 |
| Maximum (mmHg) | 0 (0) | 107 (96, 120) | 108 (97, 122) | 107 (96, 120) | < 0.001 |
| Slope[c] (mmHg/hour) | 0 (0) | -7.8 (-9.2, -6.5) | -7.3 (-8.7, -5.5) | -8 (-9.3, -6.8) | < 0.001 |
| Discharge value (mmHg) | 0 (0) | 85 (76, 95) | 85 (76, 96) | 85 (76, 95) | 0.001 |
| Discharge value ≧120 mmHg | 0 (0) | 464 (2) | 191 (3) | 273 (1.7) | < 0.001 |
| **Time from initial to last measure (hour), median (IQR)** | 0 (0) | 2.4 (1.4, 4.6) | 3.1 (1.8, 5.8) | 2.2 (1.3, 4) | < 0.001 |
| **Biochemical profiles[d], median (IQR)** | | | | | |
| Estimated Glomerular filtration rate (mL/min/1.73m2) | 4739 (20.7) | 83.4 (59.5, 98.9) | 80.1 (53.6, 96.1) | 84.8 (61.9, 100.2) | < 0.001 |
| Serum Creatinine (mg/dL) | 4739 (20.7) | 0.9 (0.7, 1.1) | 0.9 (0.7, 1.2) | 0.9 (0.7, 1.1) | < 0.001 |
| Blood urea nitrogen (mg/dL) | 6722 (29.3) | 14 (11, 20) | 15 (11, 21) | 14 (11, 19) | < 0.001 |
| Hemoglobin (g/dL) | 5412 (23.6) | 13.6 (12.2, 14.8) | 13.6 (12, 14.8) | 13.6 (12.3, 14.8) | 0.041 |
| Sodium (mmol/L) | 5866 (25.6) | 138 (136, 140) | 138 (136, 140) | 138 (136, 140) | 0.83 |
| Potassium (mmol/L) | 5636 (24.6) | 3.7 (3.4, 4) | 3.7 (3.4, 4) | 3.7 (3.4, 4) | 0.612 |
| **Medication profiles[b], n (%)** | | | | | |
| β-adrenergic antagonists | 2945 (12.9) | 3468 (17.4) | 1734 (31) | 1734 (12.1) | < 0.001 |
| Anti-platelets | 2945 (12.9) | 3006 (15.1) | 1125 (20.1) | 1881 (13.1) | < 0.001 |
| Furosemide | 2945 (12.9) | 1737 (8.7) | 742 (13.2) | 995 (6.9) | < 0.001 |
| Angiotensin- converting enzyme inhibitors | 2945 (12.9) | 3586 (18) | 2771 (49.5) | 815 (5.7) | < 0.001 |

*(Continued)*

**Table 1.** (Continued)

| Variables | Missing, n (%) | All patients | Pharmacological BP intervention | Non-pharmacological BP intervention | P-value[a] |
|---|---|---|---|---|---|
| Angiotensin receptor blockers | 2945 (12.9) | 2587 (13) | 1060 (18.9) | 1527 (10.6) | < 0.001 |
| Trichlormethiazide | 2945 (12.9) | 400 (2) | 141 (2.5) | 259 (1.8) | 0.001 |
| Calcium Channel Blockers | 2945 (12.9) | 5317 (26.6) | 3109 (55.5) | 2208 (15.4) | < 0.001 |
| α blockers | 2945 (12.9) | 1163 (5.8) | 472 (8.4) | 691 (4.8) | < 0.001 |
| Poly-anti hypertensive agents | 2945 (12.9) | 377 (1.9) | 210 (3.8) | 167 (1.2) | < 0.001 |
| Polypharmacy | 2945 (12.9) | 4546 (22.8) | 1416 (25.3) | 3130 (21.8) | < 0.001 |
| **Outcome, n (%)** | | | | | |
| **ED re-visit or inpatient service, n (%)** | | | | | |
| 7-day | 0 (0) | 2027 (8.8) | 588 (9.2) | 1439 (8.7) | 0.197 |
| 30-day | 0 (0) | 3619 (15.8) | 1024 (16.1) | 2595 (15.7) | 0.454 |
| 60-day | 0 (0) | 4567 (19.9) | 1300 (20.4) | 3267 (19.7) | 0.25 |
| **Incident stroke (after index date)[e]** | 0 (0) | 743 (3.5) | 243 (4.2) | 500 (3.3) | 0.001 |
| **Cardiovascular mortality (after index date)** | 0 (0) | 821 (3.6) | 276 (4.3) | 545 (3.3) | < 0.001 |

Abbreviations: BP, blood pressure; CMUH, China Medical University Hospital; DBP, diastolic blood pressure; ED, emergency department; eGFR indicates estimated glomerular filtration rate; EOD, end-organ damage; HTN, hypertension; IQR, interquartile range; SBP, systolic blood pressure.

[a] P-values are calculated by Kruskal-Wallis test for continuous variables and Chi-square test (or Fisher's exact test as appropriate) for categorical variables.

[b] Slope, Random slope for every person by mixed effect model.

[c] Baseline comorbidities/medication profiles that were diagnosed/taken within 1 year prior to the index date.

[d] Baseline biochemical profile that was measured within 1.5 years prior to or 0.5 years following the index date.

[e] The outcome of incident stroke exclude patients who had ever stroke before index date 1 years, and 21028 patients were left for analysis. There are 21028 patients overall, 5756 (27.4%) and 15272 (72.6%) patients in HTN Med group and Non-HTN Med group respectively.

The subgroup analysis showed that a reduction in the risk of 30-day and 60-day hospital revisits by pharmacological intervention for BP was more significant among patients who underwent any investigation for end-organ damage or those with polypharmacy (Fig 2 and S3 Table). The HRs between pharmacological intervention for BP and 30-day and 60-day hospital revisits were 0.81 and 0.95 for patients with and without polypharmacy, respectively (p for interaction <0.05, Fig 2). The HRs between pharmacological intervention for BP and 30-day and 60-day hospital revisits ranged from 0.92–0.93 for patients who underwent any investigation for end-organ damage and 0.78–0.81 for patients who were not investigated for end-organ damage (p for interaction <0.05, Fig 2). No effect modification by the a priori variables of the outcomes of incident stroke and CV mortality was noted (Figs 3 and 4 and S4 and S5 Tables). By incorporating medication histories prescribed by other hospitals using the National Health Insurance Research Database, a reduction of risk of 3- and 5-year CV mortality of 17% (95% CI, 7–25%) and 22% (95% CI, 14–28%) was demonstrated among patients receiving pharmacological intervention for high BP (S6 and S7 Tables). However, there remains no association between incident stroke and pharmacological BP control while considering patients with medication histories prescribed by other hospitals (S6 and S7 Tables). We further restricted the study population to 1281 patients with persistent hypertension above HTN-C criteria after the 4-hour ED stay and the results were similar (S8 Table).

## Discussion

Prescription of antihypertensive medication in patients who met the BP criteria for HTN-C and clinically fit to be discharged was associated with a lower risk of both 30-day and 60-day

**Table 2. Hazard ratios (HRs) with 95% confidence interval for ED revisit or inpatient admission, incident stroke, and cardiovascular mortality according to the exposure of pharmacological BP control in the ED setting.**

| Pharmacologically blood pressure reduction | N | Case | Person-year | Incidence[a] | Crude HR (95% CI) | Model 1[b] Adjusted HR (95% CI) | Model 2[c] Adjusted HR (95% CI) | Model 3[d] Adjusted HR (95% CI) |
|---|---|---|---|---|---|---|---|---|
| **ED revisit or inpatient service** | | | | | | | | |
| 7-day | | | | | | | | |
| No | 16542 | 1439 | 302.12 | 4763.0 | 1.00 (Ref) | 1.00 (Ref) | 1.00 (Ref) | 1.00 (Ref) |
| Yes | 6364 | 588 | 115.95 | 5071.2 | 1.06 (0.97–1.17) | 0.93 (0.84–1.03) | 0.89 (0.8–1) | 0.94 (0.84–1.06) |
| 30-day | | | | | | | | |
| No | 16542 | 2595 | 1212.59 | 2140.0 | 1.00 (Ref) | 1.00 (Ref) | 1.00 (Ref) | 1.00 (Ref) |
| Yes | 6364 | 1024 | 463.86 | 2207.6 | 1.03 (0.96–1.11) | 0.86 (0.8–0.94) | 0.85 (0.78–0.92) | 0.89 (0.82–0.97) |
| 60-day | | | | | | | | |
| No | 16542 | 3267 | 2329.31 | 1402.6 | 1.00 (Ref) | 1.00 (Ref) | 1.00 (Ref) | 1.00 (Ref) |
| Yes | 6364 | 1300 | 890.67 | 1459.6 | 1.04 (0.97–1.11) | 0.86 (0.8–0.93) | 0.85 (0.79–0.91) | 0.89 (0.82–0.96) |
| **Cardiovascular mortality** | | | | | | | | |
| 1-year | | | | | | | | |
| No | 16542 | 121 | 16479.43 | 7.3 | 1.00 (Ref) | 1.00 (Ref) | 1.00 (Ref) | 1.00 (Ref) |
| Yes | 6364 | 64 | 6328.24 | 10.1 | 1.38 (1.02–1.86) | 1.11 (0.81–1.54) | 0.95 (0.67–1.35) | 0.97 (0.67–1.41) |
| 3-year | | | | | | | | |
| No | 16542 | 319 | 49131.72 | 6.5 | 1.00 (Ref) | 1.00 (Ref) | 1.00 (Ref) | 1.00 (Ref) |
| Yes | 6364 | 163 | 18818.06 | 8.7 | 1.33 (1.1–1.61) | 1.07 (0.87–1.31) | 0.93 (0.75–1.16) | 0.95 (0.75–1.19) |
| 5-year | | | | | | | | |
| No | 16542 | 465 | 81416.24 | 5.7 | 1.00 (Ref) | 1.00 (Ref) | 1.00 (Ref) | 1.00 (Ref) |
| Yes | 6364 | 238 | 31140.37 | 7.6 | 1.34 (1.14–1.56) | 1.03 (0.87–1.22) | 0.89 (0.74–1.07) | 0.89 (0.74–1.08) |
| **Incident stroke** | | | | | | | | |
| 1-year | | | | | | | | |
| No | 15272 | 123 | 15193.58 | 8.1 | 1.00 (Ref) | 1.00 (Ref) | 1.00 (Ref) | 1.00 (Ref) |
| Yes | 5756 | 84 | 5708.91 | 14.7 | 1.82 (1.38–2.4) | 1.28 (0.95–1.73) | 0.91 (0.65–1.26) | 0.84 (0.59–1.19) |
| 3-year | | | | | | | | |
| No | 15272 | 280 | 45332.79 | 6.2 | 1.00 (Ref) | 1.00 (Ref) | 1.00 (Ref) | 1.00 (Ref) |
| Yes | 5756 | 166 | 16965.67 | 9.8 | 1.58 (1.31–1.92) | 1.1 (0.89–1.37) | 0.86 (0.68–1.09) | 0.84 (0.66–1.08) |
| 5-year | | | | | | | | |
| No | 15272 | 389 | 75199.98 | 5.2 | 1.00 (Ref) | 1.00 (Ref) | 1.00 (Ref) | 1.00 (Ref) |
| Yes | 5756 | 212 | 28090.76 | 7.5 | 1.46 (1.23–1.72) | 1.05 (0.87–1.27) | 0.81 (0.66–1) | 0.81 (0.65–1.01) |

[a] Incidence = No. of cases/person-years*1000.

[b] Model 1: Adjusted for age at ED admission, gender, diabetes, hypertension, cardiovascular disease, chronic kidney disease.

[c] Model 2: Further adjusted for random slope of systolic blood pressure, maximum systolic blood pressure, baseline estimated glomerular filtration rate.

[d] Model 3: Further adjusted for anti-platelet agents, polypharmacy.

BP, blood pressure; ED, emergency department.

ED revisits or hospitalizations. Regardless of the pharmacological intervention for BP, long-term adverse outcomes such as incident stroke or CV mortality were not observed. Proactive medical control of acute elevated BP was, in particular, significant. As the current study is observational, further prospective studies are warranted to confirm the potential causal role of pharmacological control of BP in hospital revisit and other clinical endpoints.

The short- and long-term outcomes of patients with HTN-C discharged from the ED are unclear in the existing literature [17]. It is worthy to note in our study population that

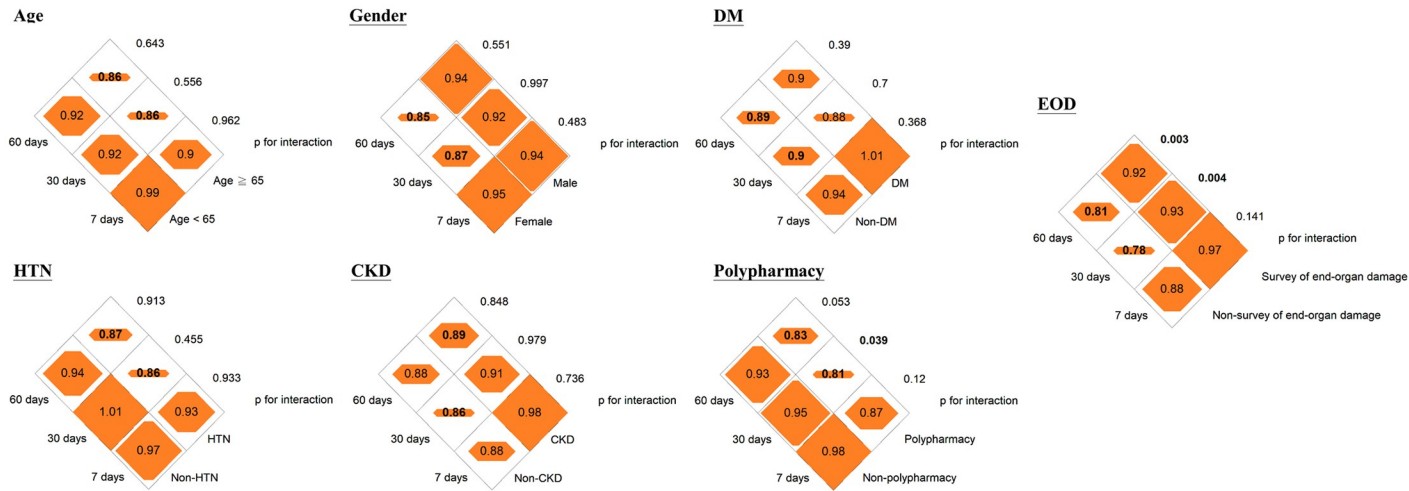

**Fig 2. Diamond graphs summarize the adjusted hazard ratios (HRs) of 7-day, 30-day, and 60-day ED revisit or inpatient admission by clinical characteristics of the study population.** DM, diabetes; HTN, hypertension; CKD, chronic kidney disease; EOD, end-organ damage. Numbers in the orange diamonds stand for the adjusted HRs, where the bold numbers represent the 95% confidence interval not overlapping with null value 1.0. Detailed information is provided in the S3A Table.

pharmacotherapy not indicated for BP control such as pain killers, antihistamines, and hypnotics were used to treat patients with HTN-C, possibly due to secondary causes in the ED setting [8]. This phenotype of HTN-C has occasionally been described as hypertensive pseudocrisis. Nonetheless, there has not been a consensus on the diagnosis of this phenomenon [18,19]. Another spectrum of HTN-C is referred to as severe symptomless hypertension (SSH); yet, whether to medically treat SSH remains controversial [20,21]. In regards to the impact on healthcare utilization, the more prominently observed protective effects of pharmacological intervention for BP on preventing revisits to the ED or inpatient admission among patients without diagnostic tests for end-organ damage imply that pharmacological control of BP may be beneficial in SSH.

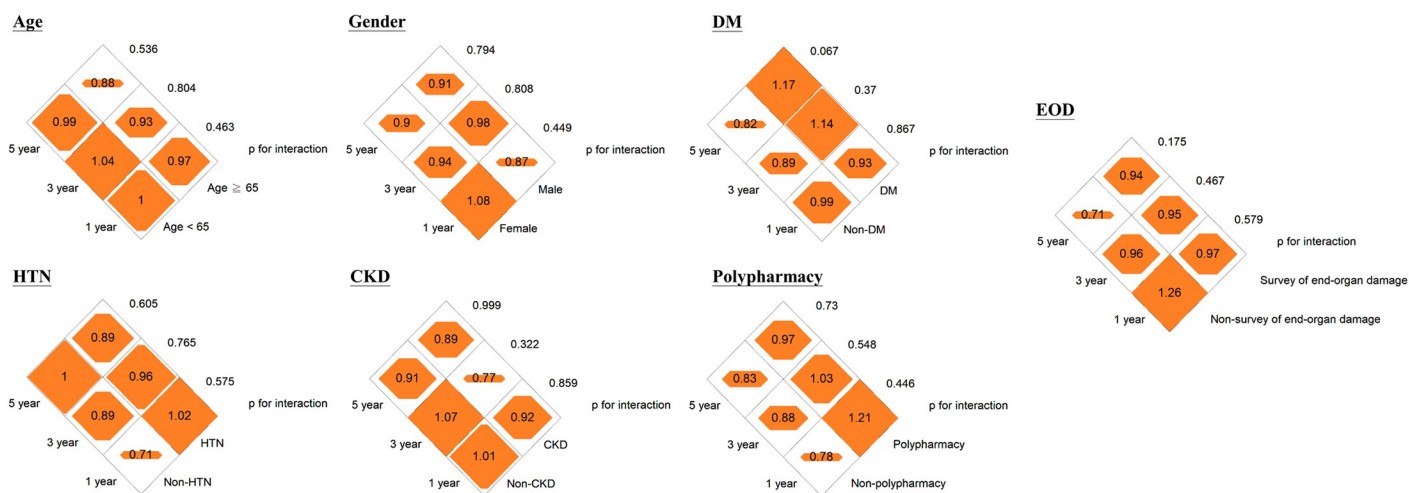

**Fig 3. Diamond graphs summarize the adjusted hazard ratios (HRs) of 7-day, 30-day, and 60-day cardiovascular mortality by clinical characteristics of the study population.** DM, diabetes; HTN, hypertension; CKD, chronic kidney disease; EOD, end-organ damage. Numbers in the orange diamonds stand for the adjusted HRs, where the bold numbers represent the 95% confidence interval not overlapping with null value 1.0. Detailed information is provided in the S3B Table.

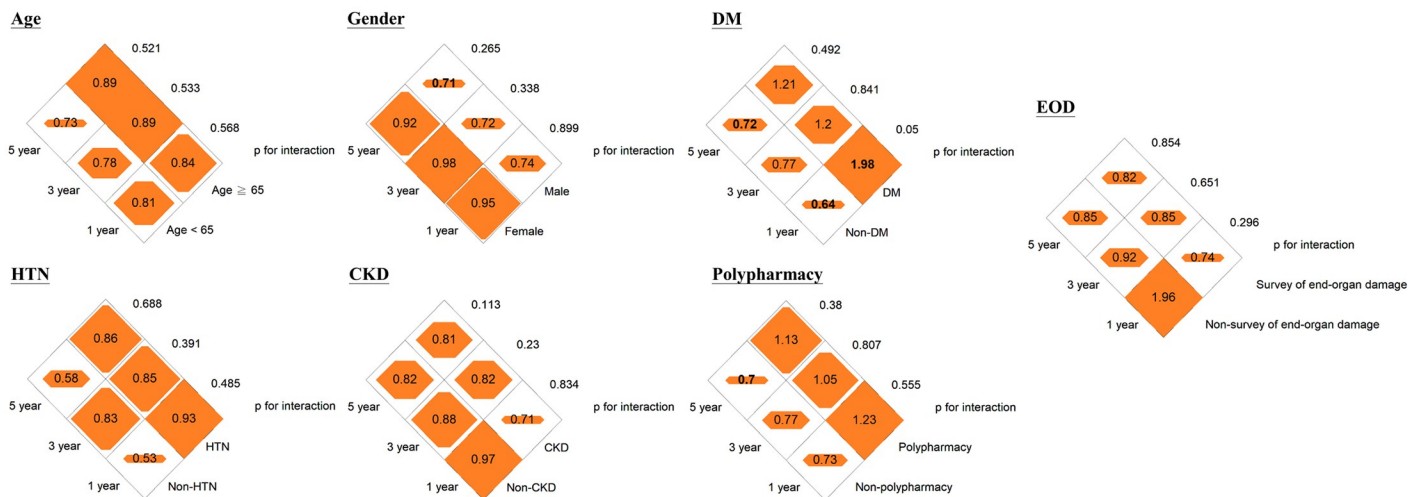

**Fig 4. Diamond graphs summarize the adjusted hazard ratios (HRs) of 7-day, 30-day, and 60-day incident stroke by clinical characteristics of the study population.** DM, diabetes; HTN, hypertension; CKD, chronic kidney disease; EOD, end-organ damage. Numbers in the orange diamonds stand for the adjusted HRs, where the bold numbers represent the 95% confidence interval not overlapping with null value 1.0. Detailed information is provided in the S3C Table.

Since 1995, Taiwan has provided universal health care coverage delivering top-quality medical care. However, the low cost of visits to an ED (less than US$ 30 including registration fee and a uniform co-payment) has turned the beneficial ED services into "convenience-store consultations" since patients find them convenient and cheap with short wait times. Between 2010 and 2016, more than 70% of 23 272 patients with HTN-C discharged from the ED did not receive any BP-lowering therapy during their stay in the ED. These patients may be allowed to stay home or visit local clinics for BP control and education. Further investigations revealed that only a fifth of them satisfied the diagnostic criteria of HTN-C at the second BP measurement. This finding implies the potential overuse of ED services, which can translate into wastage of large amounts of medical resources, as patients may receive unnecessary medical and imaging examinations. For instance, more than 14% of the discharged patients with HTN-C underwent brain CT. Our study findings demonstrate that the long-term outcomes are comparable regardless of pharmacological intervention for BP in the ED setting, therefore, the results support the clinical acumen-based decision in prescribing antihypertensive agents for patients with HTN-C who are discharged from the ED.

Our study demonstrated the need to justify the existing definitions of HTN-C. For instance, an observation period of 4 or 6 hours may be incorporated into the diagnostic criteria to separate the SSH from HTN-C. During the observational period, patients may take extra antihypertensive medications or take original medications earlier than the regular schedule and carefully record the BP trend over the observational period. A persistently high BP above the diagnostic criteria of HTN-C helps increase the diagnostic specificity. Furthermore, the diagnostic criteria for HTN-C should consider the patients' baseline medications and the burden of comorbidities as the status of polypharmacy modifies the impact of BP-lowering therapy on ED revisit or inpatient admission. Patients with special health status (such as hemodialysis) may influence the effectiveness of antihypertensive drugs in the management of hypertension [22–24]. As demonstrated by our study results, more research efforts are needed in identifying new phenotypes of this old disease first described in 1928, particularly in this big data era, which enhances the potential of continuous BP monitoring by clinicians [25,26].

The present study has several limitations. First, the observational nature of the study precludes the causal claim that clinicians can safely refrain from the pharmacological intervention for BP among patients with HTN-C who may be discharged from the ED setting. Second, case identification relied on the first BP measurement in the ED, and the chief complaints of the patients could have not been related to acute hypertension. The selection process may lead to overdiagnosis of HTN-C and potentially bias the association between pharmacological intervention for BP and our proposed clinical outcomes toward null. In addition, the discrepancies regarding the statistical significance of the associations of pharmacological BP control with 3- and 5-year CV mortality between the original single-center cohort and the same population with available medication histories in other hospitals would require more research to reconcile. Nonetheless, the present study population comprised a wide clinical spectrum from SSH to HTN-E, reflecting the real-world practice, which could strengthen the generalizability of our findings. Third, residual confounding could not be entirely excluded. Specifically, we were unable to capture information regarding drug adherence, BP control status prior to the event that required ED admission, and baseline antihypertensive medications used outside our institution. To nullify the impact of unmeasured potential confounders, we conducted E-value analysis [27,28] and the E-values ranged from 1.21–1.77 for the present study endpoints (S9 Table). It should be noted that if the strength of the potential unmeasured confounder is greater than the E-value, our findings could be affected by an unmeasured confounder. Fourth, the missing data may result in unpredictable bias in our findings. However, we performed an iterative Markov chain Monte Carlo procedure with 20 imputations and 100 iterations to replace the missing values with imputed values and the results remained robust (S10 Table).

## Conclusions

BP-lowering therapy is associated with lower risks of 30-day and 60-day ED revisits or inpatient admission, but not with short-term or long-term CV mortality or incident stroke, among patients with HTN-C who are discharged from the ED. Pharmacological intervention for BP may be particularly beneficial for hypertensive patients with polypharmacy or those who do not undergo diagnostic tests for end-organ damage. Future research efforts should focus on modifications of the old definitions of HTN-C toward better sensitivity/specificity and identify the hidden phenotypes of HTN-C by taking advantage of the advances in big medical data and increased data connectivity.

## Supporting information

**S1 Table. Type of antihypertensive drugs used for pharmacological control of blood pressure in the emergency department for hypertensive crisis (N = 6364).**
(DOCX)

**S2 Table. The International Classification of Disease (ICD) codes and medications used to define comorbidities in this study.**
(DOCX)

**S3 Table. Adjusted hazard ratios (HRs) and 95% confidence intervals of 7-day, 30-day, and 60-day ED revisit or inpatient admission by clinical characteristics of the study population.**
(DOCX)

**S4 Table. Adjusted hazard ratios (HRs) and 95% confidence intervals of 1-year, 3-year, and 5-year cardiovascular mortality by clinical characteristics of the study population.**
(DOCX)

**S5 Table. Adjusted hazard ratios (HRs) and 95% confidence intervals of 1-year, 3-year, and 5-year incident stroke by clinical characteristics of the study population.**
(DOCX)

**S6 Table. Baseline demographic and clinical characteristics of the study population verified in National Health Insurance Research Database.**
(DOCX)

**S7 Table. Hazard ratios (HRs) with 95% confidence interval for ED revisit or inpatient admission, incident stroke, and cardiovascular mortality according to the exposure of pharmacological BP control in the ED setting verified using National Health Insurance Research Database.** BP, blood pressure; ED, emergency department.
(DOCX)

**S8 Table. Hazard ratios (HRs) with 95% confidence interval for ED revisit or inpatient admission, incident stroke, and cardiovascular mortality according to the exposure of pharmacological BP control among 1281 patients with persistent high blood pressure above HTN-C criteria.** BP, blood pressure; ED, emergency department.
(DOCX)

**S9 Table. E-value analysis for ED revisit or inpatient admission, incident stroke, and cardiovascular mortality according to the exposure of pharmacological BP control in the ED setting.** BP, blood pressure; ED, emergency department.
(DOCX)

**S10 Table. Hazard ratios (HRs) with 95% confidence interval for ED revisit or inpatient admission, incident stroke, and cardiovascular mortality according to the exposure of pharmacological BP control in the ED setting based on multiple imputation data.** BP, blood pressure; ED, emergency department.
(DOCX)

## Acknowledgments

We appreciate the data exploration, statistical analysis, manuscript preparation, and the support of the iHi Clinical Research Platform from the Big Data Center of CMUH. We would like to thank the Health and Welfare Data Science Center (HWDC), Ministry of Health Welfare, and Health Data Science Center, China Medical University Hospital for providing administrative and technical support.

## Author Contributions

**Conceptualization:** Yen-Hung Liu, Chin-Chi Kuo.

**Formal analysis:** Yu-Ting Lin, Pei-Shan Chen, Shih-Ni Chang.

**Methodology:** Hsiu-Yin Chiang, Shih-Ni Chang.

**Supervision:** Chin-Chi Kuo.

**Validation:** Yen-Hung Liu, Hsiu-Chen Tsai, Chun-Hung Chen.

**Visualization:** Hsiu-Yin Chiang, Pei-Shan Chen, Shih-Ni Chang.

**Writing – review & editing:** Yu-Ting Lin, Yen-Hung Liu, Ya-Luan Hsiao, Hsiu-Yin Chiang, Chun-Hung Chen, Chin-Chi Kuo.

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
