## [Decision Letter · Decision Letter 0]

16 Feb 2021

PONE-D-21-00759

Pharmacological blood pressure control and outcomes in patients with hypertensive crisis discharged from the emergency department

PLOS ONE

Dear Dr. Kuo,

Thank you for submitting your manuscript to PLOS ONE. After careful consideration, we feel that it has merit but does not fully meet PLOS ONE’s publication criteria as it currently stands. Therefore, we invite you to submit a revised version of the manuscript that addresses the points raised during the review process.

We look forward to receiving your revised manuscript.

Kind regards,

Amjad Khan, Ph.D.

Academic Editor

PLOS ONE

Journal Requirements:

3. Please note that all PLOS journals ask authors to adhere to our policies for sharing of data and materials: https://journals.plos.org/plosone/s/data-availability. According to PLOS ONE’s Data Availability policy, we require that the minimal dataset underlying results reported in the submission must be made immediately and freely available at the time of publication. As such, please remove any instances of 'unpublished data' or 'data not shown' in your manuscript and replace these with either the relevant data (in the form of additional figures, tables or descriptive text, as appropriate), a citation to where the data can be found, or remove altogether any statements supported by data not presented in the manuscript.

Reviewers' comments:

Reviewer's Responses to Questions

**Comments to the Author**

1. Is the manuscript technically sound, and do the data support the conclusions?

Reviewer #1: Yes

Reviewer #2: Yes

2. Has the statistical analysis been performed appropriately and rigorously? 

Reviewer #1: Yes

Reviewer #2: No

3. Have the authors made all data underlying the findings in their manuscript fully available?

Reviewer #1: Yes

Reviewer #2: Yes

4. Is the manuscript presented in an intelligible fashion and written in standard English?

Reviewer #1: Yes

Reviewer #2: No

5. Review Comments to the Author

Reviewer #1: This is an interesting study which evaluates the pharmacological control of BP in terms of short- and long-term outcomes among the patients with hypertensive crises.

Overall, this is clear, concise and well-written manuscript. The authors make a systematic contribution to the research literature in this area of investigation. The introduction is relevant and theoretical in context of manuscript. Some important points have to be clarified by authors.

1. In introduction the reference indicating the prevalence of hypertensive crises in Taiwan is missing.

2. The manuscript has not discussed the proportion of hypertensive urgency and hypertensive emergencies in results separately.

3. Examination of end organ damage in addition to the mentioned diagnosis, acute pulmonary edema, and dissecting aneurysm should also be considered.

4. Table 02 at page no. 12 results should be expressed in percentages.

Few sentences from the following relevant articles can be added in Introduction or Discussion part for the improvement of the paper. Though these papers are giving information on management of hypertension in hemodialysis patients but still it can be related with control of BP in general population:

https://link.springer.com/article/10.1186/s40545-019-0169-y

https://link.springer.com/article/10.1007/s11845-018-1813-2

https://link.springer.com/article/10.1007/s40267-020-00763-5

Reviewer #2: Thank you for providing opportunity to review the draft. This is a good study and addressed important issue in ED. However, this study has few shortcomings which should be addressed before considering the draft for further processing.

1. Since substantial proportion of data is missing in the analysis, authors need to explain its impact on analysis in study limitations.

2. Authors should explain the graph and predictive model analysis in details.

3. Results may be confounded by lifestyle, compliance, dietary habits, use of alternative medicines and patient`s adherence to recommendations. How authors will nullify the impact of these confounders from the analysis?

4. I will suggest to improve the writing of manuscript. Though manuscript lack serious syntax errors but has less essence of scientific writing.

6. PLOS authors have the option to publish the peer review history of their article (what does this mean?). If published, this will include your full peer review and any attached files.

Reviewer #1: No

Reviewer #2: No

---

## [Author Response · Author response to Decision Letter 0]

20 Apr 2021

Review Comments to the Author

Reviewer #1: This is an interesting study which evaluates the pharmacological control of BP in terms of short- and long-term outcomes among the patients with hypertensive crises.

Overall, this is clear, concise and well-written manuscript. The authors make a systematic contribution to the research literature in this area of investigation. The introduction is relevant and theoretical in context of manuscript. Some important points have to be clarified by authors.

Response: We would like to thank you for your comment. We agree with and carefully follow your valuable comments to revise our manuscript accordingly. Our responses to each of your comments are provided in the point-by-point response as follows. 

1. In introduction the reference indicating the prevalence of hypertensive crises in Taiwan is missing.

Response: Thank you for this comment. However, due to we did not found the prevalence of hypertensive crises in Taiwan from public literatures, we provided the prevalence of hypertension in Taiwan and the prevalence of hypertensive crises among patients with hypertension to replace the possible prevalence of hypertensive crises in Taiwan.

Revised content:

Page 3, lines 48-50: In Taiwan, approximately one in every four Taiwanese has hypertension [4] and it is estimated that 1-2% of patients with hypertension may develop HTN-C [5]. Therefore, the management of hypertensive crisis and its associated complications are crucial in Taiwan.

2. The manuscript has not discussed the proportion of hypertensive urgency and hypertensive emergencies in results separately.

Response: Thank you. We could not obtained the real proportion of hypertensive urgency and hypertensive emergencies in this retrospective data, because not all patients were checked the end-organ damage. Moreover, due to the selection procedure used may lead to misclassification the hypertensive urgency and hypertensive emergencies, the aim of this study was explored the association between antihypertensive agents used and short- and long-term clinical outcomes among hypertensive crises rather than hypertensive urgency and hypertensive emergencies which reflecting the real-world setting. 

3. Examination of end organ damage in addition to the mentioned diagnosis, acute pulmonary edema, and dissecting aneurysm should also be considered.

Response: Thank you. We have added these two criteria in the revision text.

Revised content:

Pages 5-6, lines 112-115: Examinations for suspected end-organ damage were defined as requesting a serum creatinine exam, troponin I exam, intravenous diuretic therapy, brain CT, or aortic dissection during the index ED visits, regardless of the exam results.

4. Table 02 at page no. 12 results should be expressed in percentages.

Few sentences from the following relevant articles can be added in Introduction or Discussion part for the improvement of the paper. Though these papers are giving information on management of hypertension in hemodialysis patients but still it can be related with control of BP in general population:

https://link.springer.com/article/10.1186/s40545-019-0169-y

https://link.springer.com/article/10.1007/s11845-018-1813-2

https://link.springer.com/article/10.1007/s40267-020-00763-5

Response: Thank you for this suggestion that improve the quality of this paper. We have added the information on management of hypertension, particularly for patients with special health status. We humbly request to keep the original results of Table 2 due to the hazard ratios could be simply converted to percentages. Please further contact us if you have any concern regarding our response. Thank you! 

Revised content:

Page 16, lines 280-281: Patients with special health status (such as hemodialysis) may influence the effectiveness of antihypertensive drugs in the management of hypertension [22-24].

Reviewer #2: Thank you for providing opportunity to review the draft. This is a good study and addressed important issue in ED. However, this study has few shortcomings which should be addressed before considering the draft for further processing.

Response: We gratefully thank the reviewer for his/her thoughtful comments and suggestions. Please see below for our response to all comments and questions.

1. Since substantial proportion of data is missing in the analysis, authors need to explain its impact on analysis in study limitations.

Response: Thank you for this comment. We have added this limitation in the Discussion. Missing data may result the unpredictable bias in statistical findings. To address this problem, we performed multiple imputation by using an iterative Markov chain Monte Carlo procedure with 20 imputations and 100 iterations to replace the missing values with imputed values. The results remained robust (Table R1).

Revised content:

Page 17, lines 303-305: Fourth, the missing data may result in unpredictable bias in our findings. However, we performed an iterative Markov chain Monte Carlo procedure with 20 imputations and 100 iterations to replace the missing values with imputed values and the results remained robust (S10 Table).

Table R1. Hazard ratios (HRs) with 95% confidence interval for ED revisit or inpatient admission, incident stroke, and cardiovascular mortality according to the exposure of pharmacological BP control in the ED setting. BP, blood pressure; ED, emergency department.

Pharmacologically blood pressure reduction Model 3a Model 3

Multiple imputations

 Adjusted HR

(95% CI) Adjusted HR

(95% CI)

ED revisit or inpatient service

7-day 　 

　No 1.00 (Ref) 1.00 (Ref)

　Yes 0.94 (0.84 - 1.06) 0.96 (0.86 - 1.08)

30-day 

　No 1.00 (Ref) 1.00 (Ref)

　Yes 0.89 (0.82 - 0.97) 0.92 (0.84 - 1.00)

60-day 

　No 1.00 (Ref) 1.00 (Ref)

　Yes 0.89 (0.82 - 0.96) 0.92 (0.85 - 0.99)

Cardiovascular mortality

1-year 

　No 1.00 (Ref) 1.00 (Ref)

　Yes 0.97 (0.67 - 1.41) 1.04 (0.74 - 1.47)

3-year 

　No 1.00 (Ref) 1.00 (Ref)

　Yes 0.95 (0.75 - 1.19) 0.93 (0.75 - 1.16)

5-year 

　No 1.00 (Ref) 1.00 (Ref)

　Yes 0.89 (0.74 - 1.08) 0.93 (0.78 - 1.11)

Incident stroke

1-year 

　No 1.00 (Ref) 1.00 (Ref)

　Yes 0.84 (0.59 - 1.19) 0.92 (0.66 - 1.28)

3-year 

　No 1.00 (Ref) 1.00 (Ref)

　Yes 0.84 (0.66 - 1.08) 0.93 (0.74 - 1.16)

5-year 

　No 1.00 (Ref) 1.00 (Ref)

　Yes 0.81 (0.65 - 1.01) 0.86 (0.7 - 1.04)

aModel 3: Adjusted for age at ED admission, gender, diabetes, hypertension, cardiovascular disease, chronic kidney disease, random slope of systolic blood pressure, maximum systolic blood pressure, baseline estimated glomerular ﬁltration rate, anti-platelet agents, polypharmacy.

 

2. Authors should explain the graph and predictive model analysis in details.

Response: Thank you. We have added more details of the graph and predictive model analysis in the revised manuscript as follows:

Revised content:

Pages 6-7, lines 140-144: We also provided the visualization information of subgroup analysis to show the association between pharmacological intervention for BP among patients with HTN-C stratified by subgroup and the effect modification. We performed sensitivity analyses including patients with (1) medication histories prescribed by other hospitals and (2) persistent high blood pressure above HTN-C criteria after the 4-hour ED stay.

Page 11, lines 193-198: The HRs between pharmacological intervention for BP and 30-day and 60-day hospital revisits ranged from 0.81-0.83 for patients with polypharmacy and 0.93-0.95 for patients with no polypharmacy (p for interaction <0.05, Fig 2). The HRs between pharmacological intervention for BP and 30-day and 60-day hospital revisits ranged from 0.92-0.93 for patients who underwent any investigation for end-organ damage and 0.77-0.81 for patients who were not investigated for end-organ damage (p for interaction <0.05, Fig 2).

Page 11, lines 203-205: However, there remains no association between incident stroke and pharmacological BP control while considering patients with medication histories prescribed by other hospitals (S6 and S7 Tables).

3. Results may be confounded by lifestyle, compliance, dietary habits, use of alternative medicines and patient`s adherence to recommendations. How authors will nullify the impact of these confounders from the analysis?

Response: Thank you for this important comment. Due to the present study is a retrospective-based data, we could not obtain unmeasured confounder. Therefore, we applied E-value to address this problem [1]. The E-value is an alternative method to sensitivity analyses for unmeasured confounding factors in observational studies. If the strength of unmeasured confounding is weaker than indicated by the E-value, then the study result could not be overturned to one of the unmeasured confounder. The E-values ranged from 1.21-1.77 for the present study endpoints (Table R2). We added this information in the Discussion section as follows:

Revised content:

Page 17, lines 299-303: To nullify the impact of unmeasured potential confounders, we conducted E-value analysis [27] and the E-values ranged from 1.21-1.77 for the present study endpoints (S9 Table). It should be noted that if the strength of the potential unmeasured confounder is greater than the E-value, our findings could be affected by an unmeasured confounder.

Table R2. E-value for point estimate and confidence interval for ED revisit or inpatient admission, incident stroke, and cardiovascular mortality according to the exposure of pharmacological BP control in the ED setting. BP, blood pressure; ED, emergency department.

Pharmacologically blood pressure reduction E-value for point estimate E-value for confidence interval

ED revisit or inpatient service 

7-day 1.32 1

30-day 1.39 1.17

60-day 1.39 1.2

Cardiovascular mortality 

1-year 1.21 1

3-year 1.29 1

5-year 1.5 1

Incident stroke 

1-year 1.67 1

3-year 1.67 1

5-year 1.77 1

Reference

1. Haneuse S, VanderWeele TJ, Arterburn D. Using the E-Value to Assess the Potential Effect of Unmeasured Confounding in Observational Studies. Jama. 2019;321(6):602-3. Epub 2019/01/25. doi: 10.1001/jama.2018.21554. PubMed PMID: 30676631.

4. I will suggest to improve the writing of manuscript. Though manuscript lack serious syntax errors but has less essence of scientific writing.

Response: We thank the reviewer for the thoughtful comment on the writing of the manuscript. We have thoroughly revised and edited the sentence structures throughout the manuscript to improve flow and coherence. We have removed all language that is deemed colloquial to strengthen the manuscript.

---

## [Editor Report · Decision Letter 1]

26 Apr 2021

Pharmacological blood pressure control and outcomes in patients with hypertensive crisis discharged from the emergency department

PONE-D-21-00759R1

Dear Dr. Kuo,

We’re pleased to inform you that your manuscript has been judged scientifically suitable for publication and will be formally accepted for publication once it meets all outstanding technical requirements.

Kind regards,

Amjad Khan, Ph.D.

Academic Editor

PLOS ONE

Additional Editor Comments (optional):

All the queries and suggestions raised by the reviewers have been addressed and incorporated by the authors.
---

## [Editor Report · Acceptance letter]

9 Aug 2021

PONE-D-21-00759R1 

Pharmacological blood pressure control and outcomes in patients with hypertensive crisis discharged from the emergency department 

Dear Dr. Kuo:

I'm pleased to inform you that your manuscript has been deemed suitable for publication in PLOS ONE. Congratulations! Your manuscript is now with our production department. 

Kind regards, 

on behalf of

Dr. Amjad Khan 

Academic Editor

PLOS ONE